# Social capital, social cohesion, and health of Syrian refugee working children living in informal tented settlements in Lebanon: A cross-sectional study

**Rima. R. Habib** [1]*, **Amena El-Harakeh**[1], **Micheline Ziadee**[1], **Elio Abi Younes** [1], **Khalil El Asmar**[2]

**1** Department of Environmental Health, Faculty of Health Sciences, American University of Beirut, Beirut, Lebanon, **2** Department of Epidemiology and Population Health, Faculty of Health Sciences, American University of Beirut, Beirut, Lebanon

* rima.habib@aub.edu.lb

**Data Availability Statement:** Due to ethical issues surrounding this highly vulnerable study population

## Abstract

### Background

Since 2011, the protracted Syrian war has had tragic consequences on the lives of the Syrian people, threatening their stability, health, and well-being. The most vulnerable are children, who face interruption of schooling and child labor. This study explored the relationship between social capital and the physical health and emotional well-being of Syrian refugee working children in rural areas of Lebanon.

### Methods and findings

In this cross-sectional study, we surveyed 4,090 Syrian refugee children working in the Bekaa Valley of Lebanon in 2017. Children (8–18 years) gave direct testimony on their living and social environment in face-to-face interviews. Logistic regressions assessed the association of social capital and social cohesion with the health and emotional well-being of Syrian refugee working children; specifically, poor self-rated health, reporting a health problem, engaging in risky health behavior, feeling lonely, feeling optimistic, and being satisfied with life. Of the 4,090 working children in the study, 11% reported poor health, 16% reported having a health problem, and 13% were engaged in risky behaviors. The majority (67.5%) reported feeling lonely, while around 53% were optimistic and 59% were satisfied with life. The study findings suggest that positive social capital constructs were associated with better health. Lower levels of social cohesion (e.g., not spending time with friends) were significantly associated with poor self-rated health, reporting a physical health problem, and feeling more lonely ([adjusted odds ratio (AOR), 2.4; CI 1.76–3.36, $p < 0.001$], [AOR, 1.9; CI 1.44–2.55, $p < 0.001$], and [AOR, 0.5; CI 0.38–0.76, $p < 0.001$], respectively). Higher levels of social support (e.g., having good social relations), family social capital (e.g., discussing personal issues with parents), and neighborhood attachment (e.g., having a close friend) were all significantly associated with being more optimistic ([AOR, 1.5; CI 1.2–1.75, $p < 0.001$], [AOR, 1.3; CI 1.11–1.52, $p < 0.001$], and [AOR, 1.9; CI 1.58–2.29, $p < 0.001$],

living in a sensitive context (children and perhaps parents working with a nondocumented status, which may incur hostility), exposing their data is a potential risk to their safety and well-being. Consequently, the data from the study cannot be made public. The data underlying the results are available upon request from the Department of Environmental Health at the Faculty of Health Sciences, American University of Beirut. For inquiries about the project and data, please contact envd@aub.edu.lb.

**Funding:** RRH received funding from the International Development Research Centre (IDRC grant number 106981-001), the United Nations International Children's Emergency Fund (UNICEF), the Food and Agriculture Organization of the United Nations (FAO), and the International Labour Organization (ILO). The funders had no role in the study design, in the collection, analysis and interpretation of data, in the writing of the paper, and in the decision to submit the paper for publication.

**Competing interests:** The authors have declared that no competing interests exist.

**Abbreviations:** AOR, adjusted odds ratio; IAMP, Interagency Mapping Platform; ITS, informal tented settlement; SD, standard deviation; STROBE, Strengthening the Reporting of Observational Studies in Epidemiology.

respectively) and more satisfied with life ([AOR, 1.3; CI 1.01–1.54, $p = 0.04$], [AOR, 1.2; CI 1.01–1.4, $p = 0.04$], and [AOR, 1.3; CI 1.08–1.6, $p = 0.006$], respectively). The main limitations of this study were its cross-sectional design, as well as other design issues (using self-reported health measures, using a questionnaire that was not subject to a validation study, and giving equal weighting to all the components of the health and emotional well-being indicators).

## Conclusions

This study highlights the association between social capital, social cohesion, and refugee working children's physical and emotional health. In spite of the poor living and working conditions that Syrian refugee children experience, having a close-knit network of family and friends was associated with better health. Interventions that consider social capital dimensions might contribute to improving the health of Syrian refugee children in informal tented settlements (ITSs).

## Author summary

### Why was this study done?

- The war in Syria has resulted in a large displacement of the population to neighboring Lebanon, where many vulnerable displaced families are living in informal tented settlements (ITSs) with no access to means of livelihood and resources, and child labor among these displaced families is high.

- This study aimed to investigate the associations between social capital and cohesion and the health and emotional well-being of Syrian refugee children.

### What did the researchers do and find?

- We conducted a cross-sectional study of 1,902 households of Syrian displaced families living in ITSs in a rural area in Lebanon and interviewed 4,090 working Syrian refugee children aged between 8 and 18 years.

- Around 11% of the working children rated their health as poor, 16% reported having a physical health problem, and 13% were engaged in risky health behaviors.

- Lower social cohesion was significantly associated with reporting poor health, and a lower level of social support was significantly associated with engaging in risky health behaviors.

- Higher levels of neighborhood attachment, family social capital, and social support were significantly associated with greater optimism and life satisfaction, and higher levels of neighborhood attachment and social cohesion were significantly associated with feeling less lonely.

## What do these findings mean?

- Social capital and social cohesion are associated with reporting better health and emotional well-being among Syrian refugee children working and living in Lebanon.

- Policies aimed at promoting the physical and emotional well-being of Syrian refugee children should consider interventions that aim to maintain social capital among these populations.

## Introduction

Since 2011, the conflict in Syria has had tragic consequences on the lives of the Syrian people, threatening their stability, health, and well-being. Currently, around 6.6 million persons are internally displaced inside the country and more than 5.6 million Syrian refugees are spread across Lebanon, Turkey, Jordan, and other countries [1]. There are around 924,161 Syrian refugees in Lebanon registered with the United Nations High Commissioner for Refugees [2], but the total number of Syrian refugees residing in the country is estimated at 1.5 million [3]. These refugees are living in various urban and rural areas in Lebanon, with the highest concentration residing in the Bekaa [2], an agrarian region area located along Lebanon's eastern border with Syria.

The majority of Syrian refugees in the Bekaa are living in informal tented settlements (ITSs), which are collections of makeshift tents that provide inadequate and unsafe shelter [4]. Many such settlements are located near or on agricultural fields, which is consistent with agricultural housing practices from before the Syrian Civil War, when workers from Syria would come to tend the fields of the Bekaa on a seasonal basis [5]. Today, many of the refugees living in Bekaa still work in the agricultural sector including a high proportion of children who are employed in this field [4]. Child labor among Syrian refugees is a symptom of precarious economic conditions [6–8], functioning as a coping mechanism to deal with food insecurity and poverty [6–9]. The average monthly income for working adults is US$209 for men and US$92 for women [3]. As a result, children are being taken out of schools and pushed into work, which limits their future prospects and subjects them to health risks [7].

A recent study assessing the situation of Syrian refugees in Lebanon showed that refugee households are highly vulnerable, with around 69% below the national poverty line [3]. Factors contributing to their vulnerability include shortage in aid funds [3, 10] and restrictions on issuance of work permits for refugees [11, 12]. Children in the households have had to work to assist with the family livelihood [13]. Studies on Syrian refugees in Lebanon have focused primarily on topics related to policies and regulations, abuse, poverty, child labor, and child marriage in order to produce the necessary evidence to guide policy-making [14–17]. However, social capital and its relation to the health and well-being of refugees has not been thoroughly explored thus far. Given that Syrian refugees have been living in ITSs in Lebanon for almost 9 years now, their needs extend beyond daily necessities like food and shelter. Researchers need to give equal attention to the everyday relations that might foster resilience and contribute to well-being within this refugee community. The current study addresses this gap in the literature by exploring the relationships between social capital and cohesion and the health and emotional well-being of Syrian refugee children living and working in Lebanon.

The concept of social capital has had multiple theoretical origins, though namely from the field of sociology. Social capital has also had numerous research applications, including studies

on school performance, occupational attainment, and immigrant enterprise [18]. The meaning and conceptualization of social capital has been widely contested by scholars [19]. Commenting on the challenges that this concept poses for researchers, Schuller and colleagues [20] stated that even though there are common terms used in the literature to define social capital, these terms are operationalized in many different ways, thus undermining "the notion of social capital as a single conceptual entity." Other scholars have noted that the broadness of the processes that social capital encompasses and its definitional vagueness make its application difficult and subject to dispute [18, 21, 22]. The main aspects of social capital that are highlighted in the literature include how it is produced through interaction resulting in material and symbolic profits for members of a group [23]; its integration in everyday interactions between people in the form of obligations, expectations, information channels, and norms that encourage some practices and sanction others [24]; and how it "facilitates coordination for mutual benefit" through "features of social organization, such as networks, norms, and trust" [25].

Social capital can be measured at different levels, such as the individual, group, community, and state levels [26–28]. It can also be measured at the family level. Family social capital is identified by Coleman [24] as consisting of the relations between children and their parents or other family members, and as dependent on the quality of their relations to children [29].

Social capital has been used to study the relationship between social networks, access to resources, and health. Previous studies showed a positive association between social capital and health [30–32]. Aspects of social capital that have been linked to better health include trust (feeling safe in one's neighborhood), feelings of belonging and enjoying living in one's neighborhood [33], informal social control [34], and community participation and neighborhood connections [31]. On the other hand, social capital has been linked to the exclusion of outsiders, restriction on individual freedoms, and undermining group cohesion [18, 35].

Furthermore, social capital has been used to study children's health and well-being [36–38], and their educational performance [39, 40]. A higher level of social capital represented in family and neighborhood support has been linked to good self-rated health in adolescents [41]. Drukker and colleagues [42] found an association between children's mental health and the level of informal social control, an indicator of social capital. A recent systematic review on family social capital and health showed that this form of social capital has been associated with promoting healthy behaviors in children, protecting them from risky activities, and promoting their well-being [43]. Through their study on working children in Addis Ababa, Eriksen and Mulugeta [44] showed that children working in streets develop certain interpersonal networks (e.g., networks of siblings, relatives, and peers) that serve as social capital that can be utilized for economic gain and social protection [44].

In addition, studies have explored the relation between social capital and the health of refugee populations. In a study on different refugee populations, Loizos (2000) noted that forcibly displaced individuals work hard at protecting social capital as a way of fostering resilience [45]. Loizos [45] also condemned how state policies try to disperse refugees, thus inhibiting the formation of social capital within these communities. Social capital is vital for improving the well-being of young refugees [46], promoting community health, and preserving healthy behaviors in refugee communities [47, 48].

A conceptual model of social capital for health was developed by Carpiano (2007) based on the theoretical work by Bourdieu [49]. This model explains the influence of neighborhood social capital on health and looks at social capital as embedded within neighborhood social processes, which can in turn affect health [49]. Embedded in this model are elements of social capital, social cohesion, and health outcomes. This paper adopts Carpiano's [49] conceptualization of social capital as the interaction between a group's resources and the connections that individuals have to this group, which allow them to benefit from these resources.

This study builds on Carpiano's model [49] to critically analyze the associations of social capital and social cohesion with the health of Syrian refugee working children living in ITSs in Lebanon.

## Methods

This study is based on a cross-sectional survey conducted in 2017 with Syrian refugees living in ITSs in the Bekaa Valley of Lebanon (S1 Fig). The study explored the living and working conditions of Syrian refugee children living in these communities. The study is based on a detailed protocol that was used in designing the study (see S1 Text, Study Protocol). The detailed study methodology is also reported in recently published work [4, 6, 50, 51]. The study was approved by the Institutional Review Board at the American University of Beirut (IRB-RH1.08). This study is reported as per the Strengthening the Reporting of Observational Studies in Epidemiology (STROBE) guideline (see S1 STROBE Checklist).

We selected a random sample of 153 out of 3,748 ITSs using a database of Syrian refugees living in ITSs across Lebanon known as the Interagency Mapping Platform (IAMP) [52]. The IAMP is a database used for the coordination of humanitarian activities. It contains information such as the number of Syrian refugee ITSs, their location, and the number of tents and individuals within each settlement. In order to identify households with working children between the ages of 8 and 18 in the sampling frame, the fieldworkers coordinated with a local "Shaweesh," who is a community gatekeeper responsible for connecting migrant workers with employers [53]. Fieldworkers were recruited through a Lebanese nongovernmental organization that works with Syrian refugees in Lebanon, including those in the Bekaa, and whose activities include addressing issues related to child health and protection. All fieldworkers were native Arabic speakers. They attended a seven-day training workshop prior to beginning the fieldwork. A pilot study was carried out in an ITS that was not selected in the final study sample. This allowed for testing the questionnaire and introducing adjustments to adapt the questions to the study population.

The interviews were conducted in colloquial Arabic and took place at the participants' households with the working children. We obtained oral informed consent from the female homemaker, who was often the children's parent, and assent from the working children. If the female homemaker was not available, we interviewed an adult household member. A total of 1,907 households were contacted and 1,902 agreed to participate. The sample included 4,090 working children between 8 and 18 years of age.

## Questionnaire and measures

The questionnaire was prepared in English and then translated into colloquial Arabic (see S2 Text, Questions and Answer Choices). It was coded using KoBo Toolbox [54] and completed via electronic tablets. The questionnaire was directly administered to the working child (8–18 years) and included questions relating to social capital, social cohesion, neighborhood attachment, health behaviors, and health status.

We used several indicators that reflect the elements of Carpiano's model relating to social capital and social cohesion [49].

### Social capital measures

Social capital is conceptualized as the amount and type of resources available to a group and the connections that individuals have to this group [49]. In Carpiano's model, social capital is broken down into four components: social support used to cope with everyday life; social leverage, which serves to access information that is useful for survival and advancement;

informal social control, through which residents maintain order and security in their area; and neighborhood organization participation, which refers to collective organization for the purpose of addressing neighborhood issues [55]. In addition, we have included in our analysis another aspect of social capital relevant to children, which is family social capital that includes the quality of relations between children and their parents [29]. Indicators of social capital used in this study are as follows:

1. Social support: Participants were asked about their support network, as in, they could turn to someone (including family member, neighbor, and friend) for help when they have a personal problem. This question was dichotomized into "yes" and "no." In addition, participants were asked about the quality of their social relation. Quality of social relations combines both questions: "How is your relation with your parents?" and "How is your relation with your siblings?" Both questions were on a scale of 1 to 5, ranging from "very good" to "very poor." The sum of the two questions was calculated and scores from 1 to 4 were considered as "good" and scores from 5 to 10 as "poor."

2. Social leverage: Participants were asked whether they know of any aid organization that offers help or services to refugees. This question was dichotomized into "yes" and "no." Also, they were asked whether they go to school (yes, no) and whether they take classes outside school (yes, no).

3. Informal social control: Participants were asked how safe they feel walking alone in the street after dark. This question was dichotomized into "yes" for answers "very safe," "safe," and "no" for "somewhat safe," "not safe," and "not safe at all." This variable is used as a proxy for social control in the neighborhood.

4. Neighborhood organization participation: Participants were asked whether they are engaged in voluntary work in their neighborhood. This question was dichotomized into "yes" for answers "almost every day," "at least once a week," "once or twice a month," and "no" for "never."

In this study, family social capital was also measured to capture the effect of living within families for refugees, an important feature of refugees' living arrangements in ITSs.

Regarding family social capital, participants were asked about the quality of their relationship with their parents; this question was dichotomized into "good" for answers "very good," "good," and "average," and "poor" for "very poor" and "poor." They were also asked whether they discuss personal issues with their parents; this question was dichotomized into "yes" and "no."

## Social cohesion measures

Social cohesion was captured by participants' connectedness with their social environment. This was measured through questions on whether they spend time with friends and whether they have fun with friends; both questions were dichotomized into "yes" for answers "almost every day," "at least once a week," "once or twice a month," and "no" for "never." In addition, participants were asked whether they are cautious when dealing with other people; this question was dichotomized into "yes" for answers "strongly agree" and "agree," and "no" for "strongly disagree" and "disagree."

## Neighborhood attachment measure

Neighborhood attachment refers to how connected individuals are to their neighborhood and consequently to networks that might be a source of resources [49]. The question used for this

indicator in the study is "Do you have a close friend in the neighborhood?"; this question was dichotomized into "yes" and "no."

## Health measures

The health and emotional well-being of children was measured using a number of indicators. Self-rated health was measured by asking the child: How do you compare your health to others in your age? The five possible answers were dichotomized into "good" (including very good and good) and "poor" (including average, poor, and very poor). Children were also asked if they suffered from a health problem (Yes, No) and if they engaged in risky health behaviors (Yes, No), including having ever been a smoker and/or not taking part in sports. Participants were asked, "How do you identify yourself regarding smoking cigarettes?" Answers to the question were dichotomized into "yes" for the responses "cigarette smoker," "tried but never liked it," and "used to smoke" and into "no" for the response "non-cigarette smoker." Participants were also asked "How do you identify yourself regarding smoking waterpipe?" Answers were dichotomized into "yes" for the responses "waterpipe smoker," "tried but never liked it," and "used to smoke" and into "no" for the response "non-waterpipe smoker." Answers to the question "Do you actively participate in sports or exercise?" were dichotomized into "yes" for the responses "almost every day," "at least once a week," and "once or twice a month" and "no" for "never."

Children were also asked if they feel lonely, dichotomized into "yes" (often, sometimes) and "no" (hardly ever, never). In addition, an index on child optimism was developed using 10 questions. The 10 questions were dichotomized as "agree" and "disagree," with "agree" scored as 1 point and "disagree" as 2 points. The sum of the 10 questions was calculated, and participants who scored a value between 1 and 13 were categorized as less optimistic, while scoring between 14 and 22 was considered as being more optimistic. Moreover, child satisfaction was assessed using an index of 12 questions, which were dichotomized as "agree" and "disagree." A response of "agree" was scored as 1 point and "disagree" as 2 points. The sum of the 12 questions was calculated and participants who scored a value between 1 and 39 were categorized as more satisfied with life, while children who scored a value between 40 and 55 were considered less satisfied with life (see S2 Text, Questions and Answer Choices).

## Statistical analysis

Our defined outcome variables were as follows: self-rated health, any reported health problem, engagement in risky health behavior, loneliness, optimism, and life satisfaction. Our main exposure variables were as follows: connectedness, social support, social leverage, informal social control, neighborhood organization participation, family social capital, and neighborhood attachment.

We performed descriptive statistics of all working children between 8 and 18 years. The analysis reports included data on demographics, length of residency in Lebanon, and income. Frequencies and percentages were reported for categorical data and means and standard deviations (SDs) for continuous data. The relationships between the outcome variables and exposures were assessed using logistic regression models adjusted for district of residence, gender, and age of the household member, and adjusting for the effect of clustering at the household level. We considered an alpha value of 0.05 as statistically significant and conducted all analyses on Stata 15.0 (StataCorp, College Station, TX).

## Results

The study population consisted of 4,090 working children between the ages of 8 and 18 years. Table 1 shows their sociodemographic characteristics. The mean age of the working children

**Table 1. Sociodemographic characteristics of the working children (8–18 years) in 1,902 Syrian refugee households living in ITSs, Bekaa, Lebanon, 2017 (N = 4,090).**

| Characteristics | N | Percent |
|---|---|---|
| **Age, years** | | |
| 8–10 | 663 | 16.2 |
| 11–12 | 819 | 20 |
| 13–16 | 1,860 | 45.5 |
| 17–18 | 748 | 18.3 |
| **Gender** | | |
| Male | 2,107 | 51.5 |
| Female | 1,983 | 48.5 |
| **Field of work[a]** | | |
| Agriculture | 3,098 | 75.8 |
| Waste picking | 173 | 4.2 |
| Construction | 147 | 3.6 |
| Car wash | 100 | 2.4 |
| Street services | 98 | 2.4 |
| Factory employees | 87 | 2.1 |
| Mechanics | 78 | 1.9 |
| Other[b] | 382 | 9.3 |
| **Going to school** | | |
| Yes | 702 | 17.2 |
| No | 3,388 | 82.8 |
| **Child over age for grade[c,d]** | | |
| Yes | 589 | 83.9 |
| No | 113 | 16.1 |
| | **Mean** | **SD** |
| **Number of children/household** | 2.8 | 1.5 |
| **Hours worked daily** | 6.7 | 3 |
| **Income/month, USD** | 72.04 | 59.97 |

[a]Total greater than 100% as more than one option is possible.

[b]Includes handcrafts, packaging at markets, sewing, housekeeping, porting, and other occupations.

[c]Out of 702 children who go to school.

[d]Based on age/school grade distribution in Lebanon [56].

Abbreviations: ITS, informal tented settlement; SD, standard deviation; USD, United States dollar

was 13 years (SD = 2.7) and around 52% were males. They had been residing in Lebanon for an average of nearly 3 years (34 months). About 83% of the working children were not going to school. Out of those who were attending school (around 17%), only 16% were in grades appropriate to their age, while 84% were lagging behind in their education. Around 76% of the children worked in the agriculture sector and reported an average monthly income of US$72. On average, households housed 6.7 members, including 2.8 working children. The monthly household income per capita was US$50.70, and the majority of households (74.3%) were severely food insecure [51].

Tables 2 and 3 show associations between neighborhood and family social capital and the health of refugee working children. Two main constructs of Carpiano's model, social cohesion and social capital, were both associated with health, although the extent varied across health indicators.

**Table 2. Associations between social cohesion, social capital, and physical health for working children (8–18 years) in 1,902 Syrian refugee households living in ITSs, Bekaa, Lebanon, 2017 (N = 4,090)[a].**

| Socioeconomic characteristics | Poor self-rated health (N = 442, 10.81%) | | Reported a health problem (N = 632, 15.46%) | | Engaged in risky health behaviors (smoking/physical inactivity) (N = 535, 13.08%) | |
|---|---|---|---|---|---|---|
| | Percent (N) | AOR[b] (95% CI[c])(p-value) | Percent (N) | AOR (95% CI)(p-value) | Percent (N) | AOR (95% CI)(p-value) |
| Mean income (USD) (log) | | 0.81 (0.73–0.90) (<0.001) | | 0.7 (0.68–0.81) (<0.001) | | 1 (0.89–1.16) (0.79) |
| **Social cohesion** | | | | | | |
| *Connectedness* | | | | | | |
| Spend time with friends | | | | | | |
| -Yes (ref) | 9 (311) | 1 | 14.1 (486) | 1 | 14.2 (491) | 1 |
| -No | 20.4 (131) | 2.4 (1.76–3.36) (<0.001) | 22.7 (146) | 1.9 (1.44–2.55) (<0.001) | 6.9 (44) | 0.7 (0.47–1.08) (0.12) |
| Have fun with friends | | | | | | |
| -Yes (ref) | 8.3 (18) | 1 | 14.3 (31) | 1 | 47.5 (103) | 1 |
| -No | 11 (424) | 1.8 (0.97–3.15) (0.06) | 15.5 (601) | 1.6 (0.93–2.78) (0.09) | 11.2 (432) | 0.3 (0.18–0.38) (<0.001) |
| **Social capital** | | | | | | |
| *Social support* | | | | | | |
| Have someone to consult with on personal problems | | | | | | |
| -Yes (ref) | 10.6 (384) | 1 | 15.3 (554) | 1 | 12.4 (448) | 1 |
| -No | 12.4 (58) | 1.1 (0.77–1.64) (0.52) | 16.7 (78) | 1.3 (0.94–1.78) (0.11) | 18.6 (87) | 1.8 (1.29–2.58) (0.001) |
| *Social leverage* | | | | | | |
| Know aid organizations | | | | | | |
| -Yes (ref) | 15.5 (41) | 1 | 18.6 (49) | 1 | 29.6 (78) | 1 |
| -No | 10.5 (401) | 0.8 (0.56–1.30) (0.45) | 15.3 (583) | 1.1 (0.70–1.57) (0.82) | 12 (457) | 0.3 (0.22–0.52) (<0.001) |
| *Informal social control* | | | | | | |
| Feel safe in street after dark | | | | | | |
| -Yes (ref) | 8 (180) | 1 | 13.8 (311) | 1 | 13.5 (304) | 1 |
| -No | 14.2 (262) | 1.8 (1.39–2.30) (<0.001) | 17.4 (320) | 1.2 (0.98–1.49) (0.08) | 12.5 (230) | 1.1 (0.86–1.40) (0.46) |
| *Neighborhood organization participation* | | | | | | |
| Do volunteer work | | | | | | |
| -Yes (ref) | 10.8 (54) | 1 | 15.6 (78) | 1 | 28.5 (143) | 1 |
| -No | 10.8 (388) | 1.1 (0.74–1.54) (0.72) | 15.4 (554) | 1.2 (0.86–1.71) (0.26) | 10.9 (392) | 0.5 (0.37–0.66) (<0.001) |
| *Family social capital* | | | | | | |
| Discuss family issues with parents | | | | | | |
| -Yes (ref) | 8.2 (205) | 1 | 13.5 (335) | 1 | 10.4 (259) | 1 |
| -No | 14.7 (235) | 0.6 (0.50–0.82) (<0.001) | 18.5 (296) | 0.7 (0.55–0.85) (0.001) | 17.2 (275) | 0.6 (0.48–0.78) (<0.001) |
| **Neighborhood attachment** | | | | | | |
| Have a close friend in the neighborhood | | | | | | |
| -Yes (ref) | 10 (321) | 1 | 15 (481) | 1 | 14.4 (460) | 1 |

*(Continued)*

**Table 2.** (Continued)

| Socioeconomic characteristics | Poor self-rated health (N = 442, 10.81%) | | Reported a health problem (N = 632, 15.46%) | | Engaged in risky health behaviors (smoking/physical inactivity) (N = 535, 13.08%) | |
|---|---|---|---|---|---|---|
| | Percent (N) | AOR[b] (95% CI[c])(p-value) | Percent (N) | AOR (95% CI)(p-value) | Percent (N) | AOR (95% CI)(p-value) |
| -No | 13.7 (121) | 0.8 (0.56–1.04) (0.09) | 17.1 (151) | 0.9 (0.71–1.23) (0.65) | 8.5 (75) | 0.7 (0.49–0.99) (0.05) |

[a]Model clustered at household level and adjusted for age, gender, and district of residence.

[b]AOR—please see S1 Table for the corresponding unadjusted odds ratios.

[c]Confidence Interval

Abbreviations: AOR, adjusted odds ratio; CI, confidence interval; ITS, informal tented settlement; log, natural logarithm; ref, reference; USD, United States dollar

## Socioeconomic characteristics

The results showed that for every US$10 increase in the mean income, children were at lower odds of reporting poor self-rated health (adjusted odds ratio [AOR], 0.81; CI 0.73–0.90, $p < 0.001$) and reporting a health problem (AOR, 0.7; CI 0.68–0.81, $p < 0.001$). In contrast, for every US$10 increase in the mean income, children were at higher odds of feeling lonely (AOR, 1.11; CI 1.02–1.22, $p = 0.01$) and at lower odds of life satisfaction (AOR, 0.88; CI 0.80–0.96, $p = 0.005$).

## Social capital, social cohesion, and health

### Self-rated health and physical health problems

Around 11% of the working children rated their health as poor, 16% reported having a physical health problem, and 13% were engaged in risky health behaviors, including smoking and physical inactivity. Table 2 presents the associations between social capital and the health of working children. The results showed that lower social cohesion (not spending time with friends) was significantly associated with reporting poor health. Children who did not spend free time with their friends had higher odds of reporting poor self-rated health (AOR, 2.4; CI 1.76–3.36, $p < 0.001$) and a physical health problem (AOR, 1.9; CI 1.44–2.55, $p < 0.001$). Similarly, a lower level of informal social control, captured by not feeling safe in the street after dark, was significantly associated with poor perceived health. Children who reported feeling unsafe walking after dark were at higher odds of reporting poor self-rated health (AOR, 1.8; CI 1.39–2.30, $p < 0.001$). In addition, a lower level of family social capital (having poor relationships with the parents) was significantly associated with reporting poor health. Those who did not discuss family issues with their parents were at higher odds of reporting poor self-rated health (AOR, 0.6; CI 0.50–0.82, $p < 0.001$) and a physical health problem (AOR, 0.7; CI 0.55–0.85, $p < 0.001$), and were more likely to engage in risky behaviors (AOR, 0.6; CI 0.48–0.78, $p < 0.001$).

### Engagement in risky behaviors (smoking and physical inactivity)

Our findings also showed that a lower level of social support, measured by not having someone to consult with on personal problems, was significantly associated with engaging in risky health behaviors. Children who did not have someone to consult with were more likely to engage in risky behaviors (AOR, 1.8; CI 1.29–2.58, $p < 0.001$). On the other hand, higher levels of social leverage, neighborhood organization participation, neighborhood attachment, and social cohesion were significantly associated with less engagement in risky behaviors. Children who did not have a close friend in the neighborhood (neighborhood attachment) or did not

**Table 3.** Associations between social cohesion, social capital, and emotional well-being for working children (8–18 years) in 1,902 Syrian refugee households living in ITSs, Bekaa, Lebanon, 2017 (N = 4,090)[a].

| Socioeconomic characteristics | More lonely (N = 2,761, 67.52%) | | More optimistic (N = 2,154, 52.66%) | | More satisfied with life (N = 2,396, 58.58%) | |
|---|---|---|---|---|---|---|
| | Percent (N) | AOR[b] (95% CI[c])(p-value) | Percent (N) | AOR (95% CI)(p-value) | Percent (N) | AOR (95% CI)(p-value) |
| Mean income (USD) log | | 1.11 (1.02–1.22) (0.01) | | 0.9 (0.92–1.07) (0.83) | | 0.88 (0.80–0.96) (0.005) |
| **Social cohesion** | | | | | | |
| *Connectedness* | | | | | | |
| Spend time with friends | | | | | | |
| -No (ref) | 68.6 (2,656) | 1 | 52.7 (2,039) | 1 | 58.3 (2,258) | 1 |
| -Yes | 48.4 (105) | 0.5 (0.38–0.76) (0.001) | 53 (115) | 0.7 (0.50–0.98) (0.04) | 63.6 (138) | 1.3 (0.89–1.83) (0.18) |
| Cautious when dealing with other people | | | | | | |
| -No (ref) | 44.2 (34) | 1 | 48.1 (37) | 1 | 59.7 (46) | 1 |
| -Yes | 68 (2,727) | 3.4 (1.95–5.84) (<0.001) | 52.7 (2,116) | 1.4 (0.83–2.39) (0.21) | 58.6 (2,349) | 1.1 (0.66–1.96) (0.63) |
| **Social capital** | | | | | | |
| *Social support* | | | | | | |
| Having someone to consult with on personal problems | | | | | | |
| -No (ref) | 68.6 (321) | 1 | 37 (173) | 1 | 44 (206) | 1 |
| -Yes | 67.4 (2,440) | 0.9 (0.74–1.31) (0.94) | 54.7 (1,981) | 2.0 (1.50–2.54) (<0.001) | 60.5 (2,190) | 1.9 (1.48–2.44) (<0.001) |
| Quality of social relations | | | | | | |
| -Poor (ref) | 63 (620) | 1 | 50.5 (1,568) | 1 | 56.6 (1,757) | 1 |
| -Good | 69 (2,141) | 0.8 (0.63–0.96) (0.016) | 59.4 (586) | 1.5 (1.20–1.75) (<0.001) | 64.8 (639) | 1.3 (1.01–1.54) (0.04) |
| *Social leverage* | | | | | | |
| Going to school | | | | | | |
| -No (ref) | 70.4 (2,383) | 1 | 52.6 (1,783) | 1 | 52.4 (1,775) | 1 |
| -Yes | 53.9 (378) | 0.6 (0.48–0.79) (<0.001) | 52.9 (371) | 0.9 (0.76–1.21) (0.73) | 88.5 (621) | 6.3 (4.66–8.57) (<0.001) |
| Taking classes outside school | | | | | | |
| -No (ref) | 68 (2,725) | 1 | 52.5 (2,104) | 1 | 58.2 (2,332) | 1 |
| -Yes | 42 (34) | 0.4 (0.22–0.76) (0.005) | 59.3 (48) | 1.01 (0.57–1.78) (0.98) | 79 (64) | 1.4 (0.68–3.14) (0.33) |
| Know of aid organizations | | | | | | |
| -No (ref) | 68.8 (2,629) | 1 | 52.8 (2,017) | 1 | 58 (2,217) | 1 |
| -Yes | 49.2 (130) | 0.6 (0.39–0.81) (0.002) | 51.9 (137) | 0.9 (0.67–1.25) (0.60) | 67.8 (179) | 1.1 (0.77–1.03) (0.55) |
| *Informal social control* | | | | | | |
| Feel safe in street after dark | | | | | | |
| -No (ref) | 65 (1,196) | 1 | 52.9 (973) | 1 | 61.1 (1,125) | 1 |
| -Yes | 69.6 (1563) | 1.2 (0.96–1.38) (0.13) | 52.5 (1,180) | 0.9 (0.80–1.10) (0.44) | 56.5 (1270) | 0.9 (0.73–1.03) (0.12) |
| *Neighborhood organization participation* | | | | | | |
| Do volunteer work | | | | | | |
| -No (ref) | 68.5 (2,459) | 1 | 51 (1,829) | 1 | 58.9 (2,112) | 1 |
| -Yes | 60.3 (302) | 0.9 (0.64–1.20) (0.375) | 64.9 (325) | 1.8 (1.38–2.22) (<0.001) | 56.7 (284) | 0.7 (0.51–0.92) (0.012) |
| *Family social capital* | | | | | | |
| Discuss personal issues with parents | | | | | | |
| -No (ref) | 69.5 (1,255) | 1 | 50.1 (904) | 1 | 55.2 (997) | 1 |
| -Yes | 66.0 (1,506) | 0.9 (0.80–1.13) (0.601) | 54.7 (1,249) | 1.3 (1.11–1.52) (0.001) | 61.2 (1,398) | 1.2 (1.01–1.40) (0.04) |
| **Neighborhood attachment** | | | | | | |
| Have a close friend in the neighborhood | | | | | | |
| -No (ref) | 78.9 (697) | 1 | 38.4 (340) | 1 | 51.2 (453) | 1 |

*(Continued)*

**Table 3.** (Continued)

| Socioeconomic characteristics | More lonely (N = 2,761, 67.52%) | | More optimistic (N = 2,154, 52.66%) | | More satisfied with life (N = 2,396, 58.58%) | |
|---|---|---|---|---|---|---|
| | Percent (N) | AOR[b] (95% CI[c])(p-value) | Percent (N) | AOR (95% CI)(p-value) | Percent (N) | AOR (95% CI)(p-value) |
| -Yes | 64.4 (2,063) | 0.5 (0.39–0.62) (<0.001) | 56.6 (1,814) | 1.9 (1.58–2.29) (<0.001) | 60.6 (1,942) | 1.3 (1.08–1.60) (0.006) |

[a]Model clustered at household level and adjusted for age, gender, and district of residence.

[b]AOR—please see S2 Table for the corresponding unadjusted odds ratios.

[c]Confidence Interval.

Abbreviations: AOR, adjusted odds ratio; ITS, informal tented settlement; log, natural logarithm; ref, reference; USD, United States Dollar

engage in fun activities with friends in their free time (social cohesion) were less likely to engage in risky behaviors ([AOR, 0.7; CI 0.49–0.99, p = 0.05] and [AOR, 0.3; CI 0.19–0.39, p < 0.001], respectively) (Table 2). In addition, children who were not familiar with aid organization (social leverage) or did not engage in voluntary work (neighborhood organization participation) were less likely to engage in risky behaviors ([AOR, 0.3; CI 0.18–0.38, p < 0.001] and [AOR, 0.5; CI 0.37–0.66, p < 0.001], respectively).

## Social capital, social cohesion, and emotional well-being

### Feelings of loneliness

The majority (67.5%) of the children surveyed in this study reported feeling lonely. Around 53% were optimistic, and 59% were satisfied with life. Table 3 shows the associations between social capital, social cohesion, and emotional well-being. Results showed that higher levels of neighborhood attachment (measured by having a close friend) and social cohesion (measured by spending time with friends) were significantly associated with feeling less lonely. Children who had close friends in the neighborhood or who spent time with friends were at lower odds of feeling lonelier ([AOR, 0.5; CI 0.39–0.62, p < 0.001] and [AOR, 0.5; CI 0.38–0.76, p < 0.001], respectively). Similar results showed that higher levels of social support (having good social relations) and social leverage (knowledge of aid organization, going to school, and taking classes outside school) were significantly associated with feeling less lonely. Children who described the quality of their relations with others in their neighborhood as good or were aware of the presence of aid organization were at lower odds of feeling lonelier ([AOR, 0.8; CI 0.63–0.96, p = 0.016] and [AOR, 0.6; CI 0.39–0.81, p = 0.002], respectively). Similarly, children who went to school or took classes outside school were at lower odds of feeling lonelier ([AOR, 0.6; CI 0.48–0.79, p < 0.001] and [AOR, 0.4; CI 0.22–0.76, p = 0.005], respectively). On the other hand, lower levels of social cohesions (being cautious when dealing with others) were significantly associated with feeling lonelier. Those who reported being cautious had higher odds of feeling lonelier (AOR, 3.4; CI 1.95–5.84, p < 0.001).

### Optimism and life satisfaction

Our findings also demonstrated that higher levels of neighborhood attachment (measured by having a close friend), family social capital (measured by discussing personal issues with parents), and social support (measured by having good social relations and someone to consult with on personal problems) were significantly associated with greater optimism and life satisfaction. Children who had good social relations or had a close friend in the neighborhood were at higher odds of reporting greater optimism ([AOR, 1.5; CI 1.2–1.75, p < 0.001] and [AOR, 1.9; CI 1.58–2.29, p < 0.001], respectively), and of being more satisfied with life ([AOR,

1.3; CI 1.01–1.54, $p = 0.04$] and [AOR, 1.3; CI 1.08–1.6, $p = 0.006$], respectively). In addition, those who had someone to consult with on their personal problems or discussed their personal issues with their parents were at higher odds of reporting greater optimism ([AOR, 2; CI 1.5–2.54, $p < 0.001$] and [AOR, 1.3; CI 1.11–1.52, $p < 0.001$], respectively) and of being more satisfied with life ([AOR, 1.9; CI 1.48–2.44, $p < 0.001$] and [AOR, 1.2; CI 1.01–1.4, $p = 0.04$], respectively). Similarly, higher levels of social leverage, measured by going to school, showed a significant association with being satisfied with life. Children who went to school were at higher odds reporting greater life satisfaction (AOR, 6.3; CI 4.66–8.57, $p < 0.001$).

## Discussion

This study aimed to enhance the understanding of social capital and health by applying an adapted version of Carpiano's [49] conceptual model, which stems from Bourdieu's (1986) [23] social capital theory. This was operationalized through evaluating the association between different forms of social capital and social cohesion with the health of working children among Syrian refugees in Lebanon. Social capital and social cohesion were significantly associated with the health measures examined in this study.

Although higher income may translate to greater access to health-supporting resources and improving health, it also implies that children are spending more time at work and hence less time with friends, which can lead to greater loneliness and less life satisfaction [57].

In this study, we found that a higher level of social cohesion is associated with better health for refugee working children. For instance, refugee children in rural areas who were well connected with their neighbors and spent time with friends perceived their health as good and were less likely to feel isolated. Social cohesion is a key notion in understanding issues that revolve around the social inclusion and well-being of marginalized groups [58]. In fact, studies support the role of social connectedness (i.e., social cohesion) in promoting health, both at the community level [59] and on an individual level [60]. Immigrants who reside in neighborhoods with higher levels of collective efforts and support report better self-rated health [61], a lower extent of violence, crimes, and deprivation [62], and improved mental health [62]. Ziersch and colleagues [19] addressed this issue and highlighted the importance of a cohesive society for better mental health for refugees in rural areas.

This is complemented by the finding that being cautious and consequently less socially connected is associated with an increase in the children's loneliness and social isolation. Living in an unsafe neighborhood and in an atmosphere of impending danger represents a stressor for the children and undermines social cohesion [63–65]. This marginalized community has to be cautious due to their precarity, both in terms of poor living and working conditions [6] and the potential reality that access to emergency aid may be withdrawn at any moment [66].

Our study found an unexpected association between the risk of engagement in the unhealthy behaviors of smoking and physical inactivity with indicators of social cohesion, namely hanging out and having fun with friends. Adolescents, as part of building their social identity, engage in peer groups and are influenced by their friends [67]. Peer pressure has been highly linked to physical inactivity and smoking in previous studies [68–70]. In addition, studies have shown that being closely connected to others who engage in unhealthy behaviors such as smoking may result in a downside to health [71]. Although other studies found a positive association between higher level of social cohesion and engaging in sports activities [72], this was not the case for the refugee working children in this study, who could be too exhausted following long hours at work (an average of 6.7 hours per day) to engage in any physical activity.

In addition, the fact that many of these working children do not have access to adequate schooling is likely to affect their current and future prospects. Education is an important

component of social capital, as it is a means to access information and resources and to create better future opportunities. Moreover, an educational environment, such as a school or a learning center, can provide the space for people (both children and parents) to meet, participate in shared activities, and create relationships that promote social cohesion in a community.

Social capital is also inherent in the relationships between parents and children. Poor relationships with the parents were associated with poorer health and emotional well-being of the refugee children and lower optimism and satisfaction with life. These results confirm previous research, which showed that perceived close ties between parents and their children is a strong predictor of youth well-being [73]. Murame and colleagues [74] highlighted the importance of family in the social capital of refugees; people who had family were more likely to feel happy. According to Masten and Barnes (2018), factors associated with children's resilience include caring relationships with adults in the family and the larger community [75]. This correlates with our finding that refugees with family connection are more likely to feel optimistic. This measure (quality of parent–child relationship) has also been used by Rothon and colleagues [76] to assess the family social capital.

Neighborhood attachment, demonstrated through friendship, was an important factor associated with health. Indeed, a lower level of neighborhood attachment was associated with engaging in risky behaviors and loneliness, while being highly attached to their neighborhood showed associations with optimism and life satisfaction. Due to the threats and attacks that refugees encounter outside their communities, coming together to face these harsh conditions is essential for their survival and well-being [77]. Our findings are consistent with previous studies that highlighted the importance of peer relationships for adolescents' development and well-being [78].

As a cross-sectional study, this research cannot make causal inferences about the associations identified. Furthermore, the study had design limitations that may have affected the findings. Specifically, our survey questionnaire was not subject to a validation study. However, the questionnaire items have been commonly used in the social capital and health literature. In addition, the pilot study showed that the children understood the questions and responded to them clearly. Moreover, self-reports of health may be unreliable given the study population's limited access to adequate healthcare services that identify health problems in children. Moreover, all health conditions were given equal weighting in the construction of health indicators, not accounting for the seriousness of the health problem. A potential area of sampling bias was the focus of this study on households with working children. This likely biased the sample towards the most socioeconomically vulnerable households among the Syrian refugees in the country, thus limiting generalizability. In fact, the pervasive conditions of deprivation among responding households presented a challenge in the analysis, as many households were quite homogenous in the types of housing problems and poverty conditions they reported. In addition, the study did not use age-specific measures to capture the indicators in the study population. Despite their age range (8–18 years), the pilot study did not reveal problems in the children's understanding of the questions. The working children in this marginalized community have taken responsibilities at an early age and have had a wider exposure to the issues addressed in this study than what can be expected of children in their age.

In this study, we observed interconnections between social capital, social cohesion, and health. The social capital elements in this study were associated with refugee working children's health. We also highlight the potential role of social cohesion in establishing and maintaining social connectedness that is related to positive health. The findings also highlight the relationship of social capital with belonging and social inclusion. Although the associations between social capital and health have been well discussed in the literature, there is a dearth of research on the interplay between these factors among Syrian refugees in humanitarian

settings. This paper focused on the interconnectedness between social capital dimensions and the health of Syrian refugee children. These relationships have not been well addressed among Syrian refugee children facing precarious living and working conditions, which is a main contribution of this study.

Considerations of social capital as a public policy tool to achieve social cohesion need to take into account the alternative conceptions of social capital rooted in an understanding of the refugees' contexts, displacement, and social interactions. In addition, several upstream factors such as the role of neighborhoods and communities, role of institutions, and social and economic policies need to be considered to reduce the burden on children, enable engagement in schooling, and address the likely intergenerational impacts of poverty for the refugee children and their families.

Interventions that consider social capital dimensions might contribute to improving the health of Syrian refugee children in ITSs. Such interventions could include setting up peer-to-peer support programs that promote emotional well-being among child refugees. Other initiatives include the creation of spaces where children can play and socialize, enhancing social connectedness in the community. In addition, organizing social gatherings that bring together Syrian refugees and members of the host communities can promote more interaction, cooperation, and solidarity for the refugee children and their families.

## Conclusions

This study highlights the association between social capital, social cohesion, and refugee working children's physical and emotional health. In spite of the poor living and working conditions that Syrian refugee children encounter, having a close-knit network of family and friends was associated with better health.

## Supporting information

**S1 Fig. Selected ITSs in the Bekaa, Lebanon [51].** ITS, informal tented settlement
(TIFF)

**S1 Text. Study Protocol.**
(DOCX)

**S2 Text. Questions and answer choices.**
(DOCX)

**S1 STROBE Checklist. STROBE, strengthening the reporting of observational studies in epidemiology**
(DOC)

**S1 Table. Unadjusted odds ratios for the associations between social cohesion, social capital, and physical health for working children (8–18 years).**
(DOCX)

**S2 Table. Unadjusted odds ratios for the associations between social cohesion, social capital, and emotional well-being for working children (8–18 years).**
(DOCX)

## Acknowledgments

The authors thank the Syrian refugees in Lebanon who agreed to participate in this project. We also thank all those who provided support on this project.

## Author Contributions

**Conceptualization:** Rima. R. Habib.

**Formal analysis:** Elio Abi Younes, Khalil El Asmar.

**Funding acquisition:** Rima. R. Habib.

**Investigation:** Rima. R. Habib.

**Methodology:** Rima. R. Habib.

**Project administration:** Rima. R. Habib.

**Resources:** Rima. R. Habib.

**Supervision:** Rima. R. Habib.

**Validation:** Rima. R. Habib.

**Visualization:** Rima. R. Habib.

**Writing – original draft:** Rima. R. Habib, Amena El-Harakeh, Micheline Ziadee.

**Writing – review & editing:** Rima. R. Habib, Amena El-Harakeh, Micheline Ziadee, Elio Abi Younes.

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
