## [Decision Letter · Decision Letter 0]

13 Apr 2020

Dear Dr. Habib,

Thank you very much for submitting your manuscript "Social Capital, Social Cohesion, and Health of Syrian Refugee Working Children Living in Informal Tented Settlements in Lebanon" (PMEDICINE-D-19-03762) for consideration in PLOS Medicine's Special Issue on Refugee and Migrant Health.

Your paper was evaluated by a senior editor and discussed among all the editors here. It was also discussed with the Guest Editors, and sent to three independent reviewers, including a statistical reviewer. The reviews are appended at the bottom of this email and any accompanying reviewer attachments can be seen via the link below:

[LINK]

In light of these reviews, I am afraid that we will not be able to accept the manuscript for publication in the journal in its current form, but we would like to consider a revised version that addresses the reviewers' and editors' comments. Obviously we cannot make any decision about publication until we have seen the revised manuscript and your response, and we plan to seek re-review by one or more of the reviewers. 

In revising the manuscript for further consideration, your revisions should address the specific points made by each reviewer and the editors. Importantly, please be sure to address the critique of the statistical reviewer (Reviewer 1) and please be sure to update the citation information for reference 51. Also, please describe in your response letter specifically how this new study represents a novel advance beyond the study reported in reference 51.

Please also check the guidelines for revised papers at http://journals.plos.org/plosmedicine/s/revising-your-manuscript for any that apply to your paper. In your rebuttal letter you should indicate your response to the reviewers' and editors' comments, the changes you have made in the manuscript, and include either an excerpt of the revised text or the location (eg: page and line number) where each change can be found. Please submit a clean version of the paper as the main article file; a version with changes marked should be uploaded as a marked up manuscript.

We expect to receive your revised manuscript by May 04 2020 11:59PM. Please email us (plosmedicine@plos.org) if you have any questions or concerns.

We look forward to receiving your revised manuscript. 

Sincerely,

Caitlin Moyer, Ph.D.

Associate Editor 

PLOS Medicine

plosmedicine.org

1.Data Availability Statement: The Data Availability Statement (DAS) requires revision. Thanks for noting that the data are contained within the manuscript. PLOS Medicine requires that the de-identified data underlying the specific results in a published article be made available, without restrictions on access, in a public repository or as Supporting Information at the time of article publication, provided it is legal and ethical to do so. Please see the policy at 

http://journals.plos.org/plosmedicine/s/data-availability

and FAQs at 

http://journals.plos.org/plosmedicine/s/data-availability#loc-faqs-for-data-policy

Specifically, for each data source used in your study: 

2. Title: Please revise your title according to PLOS Medicine's style. Please place the study design ("A randomized controlled trial," "A retrospective study," "A modelling study," etc.) in the subtitle (ie, after a colon).

3. Abstract: Methods and Findings: Please clearly state the main outcome measures of interest in the study.

4. Abstract: Methods and Findings: Please quantify the main results for the reported associations with both 95% CIs and p values. Please define the abbreviation AOR at first use.

5. Abstract: Methods and Findings: Please include the important variables that are adjusted for in the analyses.

6. Abstract: Methods and Findings: In the last sentence of the Abstract Methods and Findings section, please describe the main limitation(s) of the study's methodology.

7. Abstract: Conclusions: Please address the study implications without overreaching what can be concluded from the data; please revise the sentence "Multidimensional interventions that consider social capital dimensions are needed to improve the health of Syrian refugee children in informal tented settlements.” as it suggests a causal relationship between social capital and health, and your study cannot speak to a causal relationship.

8. Author Summary: At this stage, we ask that you include a short, non-technical Author Summary of your research to make findings accessible to a wide audience that includes both scientists and non-scientists. The Author Summary should immediately follow the Abstract in your revised manuscript. This text is subject to editorial change and should be distinct from the scientific abstract. Please see our author guidelines for more information: https://journals.plos.org/plosmedicine/s/revising-your-manuscript#loc-author-summary

9. Introduction, paragraph 3 (and throughout): Your study is observational and therefore causality cannot be inferred. Please remove language that implies causality, such as “The current study addresses this gap in the literature by exploring social capital and social cohesion and their effects on the health of Syrian refugee children living and working in Lebanon.”

We suggest: “The current study addresses this gap in the literature by exploring the relationships between social capital and cohesion and the health of Syrian refugee children living and working in Lebanon.” or similar.

10. Methods: Questionnaire: Please include the questionnaire and all the questions relating to outcome measures (e.g. social capital, social cohesion, neighborhood attachment, health behaviors, and health status), as well as a description of how answers were quantified, as a supporting information file, and refer to it in the text where the questionnaire is described.

11. Results: Socioeconomic characteristics paragraph: Please provide the p values in addition to the 95% CIs associated with these analyses.

12. Results: Self-rated health and physical health problems paragraph: Please provide the p values in addition to the 95% CIs associated with these analyses.

13. Results: Engagement in risky behaviors (smoking and physical inactivity): Please provide the p values in addition to the 95% CIs associated with these analyses.

14. Results: Feelings of loneliness paragraph: Please provide the p values in addition to the 95% CIs associated with these analyses.

15. Results: Optimism and life satisfaction paragraph: Please provide the p values in addition to the 95% CIs associated with these analyses.

16. Discussion, paragraph 6: Please revise the sentence as it implies causality: “Poor relationships with the parents affected the health and wellbeing of the refugee children and reduced their optimism and satisfaction with life.” We suggest: “Poor relationships with the parents were associated with poorer health and wellbeing of the refugee children and lower optimism and satisfaction with life.” or similar.

17. Discussion, Implications paragraph: Throughout this paragraph, please address the study implications without overreaching what can be concluded from the data; the phrase "In this study, we observed ..." may be useful. Please revise or remove the sentence “The studied social capital elements represented important factors in predicting a refugee working child’s health.” as your results cannot speak to the predictive relationship between these elements, only that they were associated. Please similarly revise the rest of the paragraph to reduce causal implications of your study.

18. Table 1: It is confusing that standard deviation is presented in the percent column for some variables. In the legend, please define abbreviations for SD and USD.

19. Table 2 and Table 3: Please provide the p values for these comparisons. Please provide the results from unadjusted analyses as well as the adjusted comparisons, and in the Table legend please define the abbreviations for AOR, CI and USD.

20. Checklist: Please ensure that the study is reported according to the STROBE guideline, and include the completed STROBE checklist as Supporting Information. When completing the checklist, please use section and paragraph numbers, rather than page numbers. 

Please add the following statement, or similar, to the Methods: "This study is reported as per the Strengthening the Reporting of Observational Studies in Epidemiology (STROBE) guideline (S1 Checklist)."

21. References: RE reference 51 listed as under review, papers cannot be listed in the reference list until they have been accepted for publication or are otherwise publicly accessible (for example, in a preprint archive). Please update reference 51. Please also in your response, note the advance that the current study provides over the study published in ref 51.

Comments from the reviewers:

Reviewer #1: This is an interesting study on the association between social capital/cohesion and health of Syrian refugee working children in Lebanon. The dataset, statistical method and analysis, presentation (tables and figures) and interpretation of results are mostly adequate. The descriptive part of study is useful. However, there are a few major issues needing attention.

1) Although based on the conceptual model of social capital for health by Carpiano (2007), the survey questionnaire on social capital and cohesion has not been validated by any means therefore subject to scrutiny, which led to validity and reliability issues for the study.

2) The cross-section design of the study cannot offer any evidence on the causal relationships as hinted by authors in the abstract "those who enjoy a closely-knit network of family and friends have experienced better health than those who did not". It could be either way, low social networks led to poor health or poor health led to low social networks. Authors need to be very clear and strict on this.

3) All the associations discovered in the study seem well-known already. Not sure what is the novelty of the study.

Reviewer #2: Thank you for the opportunity to review this manuscript. This study investigates how social capital and cohesion is associated with physical and emotional health of refugee children in displacement. 

Overall, the manuscript is well organized and clearly written. The topic is important in understanding resilience and protective factors for overall health among refugee children in LMICs. Below are some suggestions for better clarity and readability. 

p. 3: Many potential readers may not be familiar with the geographic setting. Authors may consider using a map (that in relation to sampling using IAMP might be even better). 

Syrian refugees' livelihood might need more explanation as most refugees are not allowed to work in other contexts and (Syrian) refugees in Lebanon are in unique situations (e.g. not being called a "refugee" but "migrant"), which should be discussed for better contextual information. 

p.4: The way authors introduced social capital and social cohesion is quite abrupt. It would be nice to put more background and justification for introducing these concepts. 

P.5: The authors' own definition of social capital (based on the adopted conceptual model) should be demonstrated around here. The first paragraph of p.5 describes social cohesion as part of social capital and yet the figure separates them as distinct concepts, which is confusing. 

pp.6-9: How did the study deal with social desirability issues? The health measures include sensitive questions and the fieldworkers might happened to be acquaintances to those participants. Also, the age range is quite wide for some of these questions in avoiding measurement errors (e.g. significant gaps in capability and relevancy between nine and eighteen-year-old children in reporting loneliness and/or risky health behavior). More justification is necessary for adopting these questions. 

Throughout the findings, authors should clearly mention any controlling factors in the regression models in text. Especially, age and possibly gender would affect the main outcomes substantially, so the results in Tables 2&3 might consider including these controlled demographic factors. 

P.20: The results in the second paragraph is something age-sensitive and yet authors haven't explained this factor at all and treated age as if something neutral and insignificant. Another main flaw of the study is too narrowly defined and assessed "family social capital", which is defined too individualistically and not accurately reflecting family-level social capital. 

Limitations can be more expanded with some of the above-mentioned comments. Implications can also include more specific suggestions for policy and practice on the ground. 

Reviewer #3: This nicely written manuscript contributes to the growing body of literature examining the health, well-being and resilience of families and children of refugee background in transit and settlement countries. The intergenerational implications of fleeing war and human rights abuses is a critical global public health issue.

The study context and concepts of social capital and social cohesion are well described in the introduction and the discussion section is thoughtful. The authors may wish to consider the following in revising their manuscript.

Abstract

1. Please include age of the children.

Methods

2. Two references to the study authors 'recently published work' (page 6)are the same/listed twice in the reference list (reference 4 & reference 52).

3. What information is contained in the Interagency Mapping Platform as the database used for sampling?

4. Can one assume that you interviewed all children that met the eligibility criteria in the household (with consent)? Reporting the number of children per household would be helpful to provide some context for the analysis where you have adjusted for effect of clustering at household level.

5. What languages were the interviews conducted in? Was information collected on the literacy levels of the children? Completed schooling appropriate to their age? Please detail all socio-demographic data collected.

6. Were fieldworkers recruited on the basis of their experience of working/research with young people and/or their language skills?

7. The detailed list of questionnaire items and measures in the text could be complemented by including the entire interview schedule/questionnaire in an appendix.

8. Was any piloting of the interview questions undertaken?

Results

9. Table 1: age, hours worked daily and income (mean, standard deviation) require different column headings than n,%

10. This reviewer's preference is to see the unadjusted Odds Ratios presented; these could be included in tables as supplementary files.

11. It is best to avoid repeating the AdjOR and CI values in the text when these are detailed in the tables. 

12. Typo at the bottom of Table 3 (page 19) with the number '40'?

Discussion

13. Typo page 20, line 49 'spending' instead of 'sending'

14. Reference is made on page 22 to the challenge in the analysis in terms of homogeneity of households in terms of housing problems and poverty conditions. Some details earlier about household composition, household income, housing and food insecurity would be useful in placing this limitation into context.

15. The authors have an opportunity here to articulate what the potential multidimensional interventions could be. What role do (or can) existing social, humanitarian, health service play in innovative approaches to strengthening the social capital and cohesion of working children in these and other settlements? Did the authors hear from the participating children as to their perspectives about what strategies would help them?

[LINK]

---

## [Decision Letter · Decision Letter 1]

2 Jul 2020

Dear Dr. Habib,

Thank you very much for re-submitting your manuscript "Social Capital, Social Cohesion and Health of Syrian Refugee Working Children Living in Informal Tented Settlements in Lebanon: A Cross-Sectional Study" (PMEDICINE-D-19-03762R1) for consideration in PLOS Medicine's Special Issue on Refugee and Migrant Health.

I have discussed the paper with my colleagues and the guest editor and it was also seen again by two reviewers. I am pleased to say that provided the remaining reviewer points and editorial and production issues are dealt with we are planning to accept the paper for publication in the journal.

[LINK]

We look forward to receiving the revised manuscript by Jul 09 2020 11:59PM. 

Sincerely,

Caitlin Moyer, Ph.D.

Associate Editor 

PLOS Medicine

plosmedicine.org

Requests from Editors:

1.Reviewer 1 comment: Please do include your response regarding the questionnaire validity as a limitation of the study, in the Discussion section and in the limitations of the Abstract.

2.Reviewer 1 comment: Please do incorporate your response to the reviewer regarding the study's novelty into the discussion.

3.Reviewer 3 comments: Please do address reviewer 3’s comments where appropriate in the Discussion section.

4.Data availability statement: Please revise your statement and include data access and contact information. You have noted in your response that “For enquiries about the project and data, please contact Ms. H. Mansour at the Faculty of Health Sciences, American University of Beirut. Email: hm102@aub.edu.lb.” However, please note that the contact point for granting data access cannot be one of the study’s authors. 

You also note that “All relevant data are within the manuscript and its Supporting Information files.” PLOS defines the “minimal data set” to consist of the data set used to reach the conclusions drawn in the manuscript with related metadata and methods, and any additional data required to replicate the reported study findings in their entirety. Authors do not need to submit their entire data set, or the raw data collected during an investigation, but please submit the values behind the means, standard deviations and other measures reported.

5.Prospective analysis plan: Did your study have a prospective protocol or analysis plan? Please state this (either way) early in the Methods section.

6.Abstract: Methods and Findings: Please change “(>8 to 18 years)” to “(8 to 18 years)” for the ages of the children.

7.Abstract: Methods and Findings: For the associations between family social capital and social support and reporting more satisfaction with life, please report exact p values rather than p<0.04.

8.Author summary: Why was this study done?: Please revise to reduce the number of bullet points, as follows:

--The war in Syria has resulted in a large displacement of the population to neighboring

Lebanon, where many vulnerable displaced families are living in informal tented settlements with no access to means of livelihood and resources and child labor among these displaced families is high.

--This study aimed to investigate the associations between social capital and cohesion and

the health and emotional well-being of Syrian refugee children.

9.Author summary: What did the researchers do and find?: Please revise to reduce the number of bullet points, as follows:

--We conducted a cross-sectional study of 1,902 households of Syrian displaced families

living in informal tented settlements in a rural area in Lebanon, and interviewed 4,090 working Syrian refugee children aged between 8 and 18 years.

--Around 11% of the working children rated their health as poor, 16% reported having a

physical health problem, and 13% were engaged in risky health behaviors.

--Lower social cohesion was significantly associated with reporting poor health, and a lower level of social support was significantly associated with engaging in risky health behaviors.

--Higher levels of neighborhood attachment, family social capital, and social support were significantly associated with greater optimism and life satisfaction, and higher levels of neighborhood attachment and social cohesion were significantly associated with feeling less lonely.

10.Introduction: Please remove section sub-headers from the Introduction.

11.Introduction: bottom of page 5: Instead of bullet points, please summarize the definition of social capital in the most relevant way for your study (in paragraph form).

12.Methods: First paragraph: Please remove the strike-through text for citation 52.

13.Methods: Please specify the nature of informed consent, and can you please clarify if the “female homemaker” was the children’s parent or guardian (or other relationship).

14.Methods: top of page 10: Please clarify what is meant by SC15, and if an abbreviation please spell it out in the text somewhere “‘Do you have a close friend in the neighborhood?” (SC15)”

15.Methods: Statistical analysis: Please mention the factors adjusted for in your analyses.

16.Results: first paragraph, page 12: The following sentence seems speculative, and might be more appropriate in the discussion, unless you are presenting supporting data in the paper: “This lag in age to school grade may have been the consequence of the children’s engagement in work activities.”

17.Results page 14: Socioeconomic characteristics: for the relationships between income and health and loneliness measures, please also include the unadjusted OR, or note the table where they are presented.

18.Results: page 18: In the section on Optimism and Life Satisfaction, please report the exact p value for the relationship between social relations and satisfaction with life, rather than “p<0.04”. 

19.Discussion: first paragraph: Please replace the word “strongly” with “significantly” if that is your intended meaning. Otherwise, please delete the word “strongly” as it is a subjective term.

20.Discussion: third paragraph: Please revise the first sentence of this paragraph to: “In this study, we found that a higher level of social cohesion is associated with better health for refugee working children” or similar.

21.Discussion: Middle of page 24: Please revise the following sentences to: “Indeed, a lower level of neighborhood attachment was associated with engaging in risky behaviors and loneliness, while being highly attached to their neighborhood showed associations with optimism and life satisfaction.” or similar, to clarify.

22.Discussion: Please remove the sub-headings “Limitations” and “Implications”

23.Discussion: As mentioned by the reviewer, on page 25 please clarify the sentence “The working children in this marginalized community have taken responsibilities at an early age, and tend to be more mature for their age.” indicating what is meant by “more mature” and any supporting references.

24.Discussion: Please revise the following sentences in the Implications paragraph to avoid implying causality. We suggest: “We also highlight the potential role of social cohesion in establishing and maintaining social connectedness that is related to positive health. The findings also highlight the relationship of social capital with belonging and social inclusion.”

25.Discussion: Conclusions: Please revise the following sentence in the Conclusion paragraph: “In spite of the poor living and working conditions that Syrian refugee children encounter, those who enjoy a closely-knit network of family and friends reported better health than those who did not.”

26.Discussion: Conclusions: Please keep the conclusions focused on the main points of the study’s findings. The following sentences seem more appropriate for the paragraph discussing implications and future directions: “Multidimensional interventions that consider social capital dimensions might contribute to improving the health of Syrian refugee children in informal tented settlements. Such interventions could include setting up peer-to-peer support programs that promote emotional wellbeing among child refugees. Other initiatives include the creation of spaces where children can play and socialize, enhancing social connectedness in the community. In addition, organizing social gatherings that bring together Syrian refugees and members of the host communities can promote more interaction, cooperation, and solidarity for the refugee children and their families.”

27.Table 1: Please remove the strikethrough text.

28.Figure 1: Is this figure taken directly from Carpiano, 2007?- if so we may not be able to include it due to copyright issues.

29.S1 Figure (map): Please confirm that you are able to freely share this image under CC BY 4.0 (In particular, given the text from https://scholarworks.aub.edu.lb/handle/10938/21507: “Copyright Statement

All digitized texts and images in the AUB Libraries collections are for the personal, not-for-profit use of students, scholars, and the public. Any such use must name The American University of Beirut Libraries as the original source for the material. All texts and images are subject to copyright laws and, except where noted otherwise, are the property of the University Libraries. Commercial use, print or electronic re-publication of text or images (including reposting on the web an integral text or image) is strictly prohibited without prior written permission from the University Libraries. Reproduction Service Fee information is available on our website.”)

30.S4 file (supporting information Table 2 and Table 3): Please include descriptive legends for both tables, and please define abbreviations for USD, CI, and OR within the legend.

31.STROBE checklist: Thank you for including the checklist. There are two question marks that need to be replaced with location information for reporting participant numbers and unadjusted results, under “Participants” and “Main Results”.

Comments from Reviewers:

Reviewer #1: Thanks authors for their effort to improve the manuscript. I am mostly satisfied with the response and the revision. However, the response to my question on the validity of the questionnaire (comment 1) should be included in the limitation in the discussion of the paper. Also, the response to the novelty of the study (comment 3) should be included and highlighted in the discussion. This is because both questions are very important and of interesting to all the readers.

Reviewer #3: Thank you to the authors for their careful and thoughtful response to the reviewers' suggestions. I have two remaining comments:

1. In response the age range of the children and capacity of the younger children to respond to the questions, the authors note that the children are mature for their age. I would be somewhat cautious about making this statement as maturity can be considered as a multi-dimensional concept. It is striking the proportion of young working children including those as young as eight; and the level of minimal/no schooling. The finding in relation to schooling should be highlighted, and a comment made on how this may relate to social capital and social cohesion. 

2. Thank you for the inclusion of the paragraph of how these findings could be used to inform innovation to build and support social capital in these communities. I did wonder if the authors could consider going beyond this by commenting on the need to focus on 'up-stream' factors to reduce the burden on children, enable engagement in schooling, and address the likely intergenerational impacts of poverty for these refugee children and their families.

[LINK]

---

## [Editor Report · Decision Letter 2]

21 Jul 2020

Dear Dr Habib, 

On behalf of my colleagues and the academic editor, Dr. Paul Spiegel, I am delighted to inform you that your manuscript entitled "Social Capital, Social Cohesion and Health of Syrian Refugee Working Children Living in Informal Tented Settlements in Lebanon: A Cross-Sectional Study" (PMEDICINE-D-19-03762R2) has been accepted for publication in PLOS Medicine. 

PRODUCTION PROCESS

PRESS

PROFILE INFORMATION

Thank you again for submitting the manuscript to PLOS Medicine. We look forward to publishing it. 

Best wishes, 

Caitlin Moyer, Ph.D.

Associate Editor 

PLOS Medicine

plosmedicine.org